# Cortactin in Lung Cell Function and Disease

**DOI:** 10.3390/ijms23094606

**Published:** 2022-04-21

**Authors:** Mounica Bandela, Patrick Belvitch, Joe G. N. Garcia, Steven M. Dudek

**Affiliations:** 1Department of Biomedical Engineering, College of Engineering, University of Illinois at Chicago, Chicago, IL 60607, USA; mbande4@uic.edu; 2Division of Pulmonary, Critical Care, Sleep and Allergy, Department of Medicine, University of Illinois at Chicago, Chicago, IL 60612, USA; pbelvitc@uic.edu; 3Department of Medicine, University of Arizona, Tucson, AZ 85721, USA; skipgarcia@arizona.edu

**Keywords:** cortactin, endothelium, actin cytoskeleton, ARDS, COPD, asthma

## Abstract

Cortactin (CTTN) is an actin-binding and cytoskeletal protein that is found in abundance in the cell cortex and other peripheral structures of most cell types. It was initially described as a target for Src-mediated phosphorylation at several tyrosine sites within CTTN, and post-translational modifications at these tyrosine sites are a primary regulator of its function. CTTN participates in multiple cellular functions that require cytoskeletal rearrangement, including lamellipodia formation, cell migration, invasion, and various other processes dependent upon the cell type involved. The role of CTTN in vascular endothelial cells is particularly important for promoting barrier integrity and inhibiting vascular permeability and tissue edema. To mediate its functional effects, CTTN undergoes multiple post-translational modifications and interacts with numerous other proteins to alter cytoskeletal structures and signaling mechanisms. In the present review, we briefly describe CTTN structure, post-translational modifications, and protein binding partners and then focus on its role in regulating cellular processes and well-established functional mechanisms, primarily in vascular endothelial cells and disease models. We then provide insights into how CTTN function affects the pathophysiology of multiple lung disorders, including acute lung injury syndromes, COPD, and asthma.

## 1. Introduction

### 1.1. Structure and Function of Cortactin

In eukaryotic cells, the cytoskeleton provides mechanical support that is essential for normal cell function. The cytoskeleton consists of three main components: microtubules, intermediate filaments, and actin filaments [1]. These structural elements maintain cell shape and carry out necessary functions such as migration and cell division. Actin filaments are abundant in all eukaryotic cells, are highly concentrated in certain cell structures such as the cell cortex and are critical to cytokinesis and cell movement [1,2]. Actin filaments are formed by polymerization in an ATP-dependent process which includes three phases, nucleation, elongation, and steady state, and provides the driving force necessary for several cellular functions [3]. The Arp2/3 complex promotes the formation of branched actin networks and also interacts with tyrosine kinases and other actin-binding proteins regulating actin polymerization [4]. For example, members of the Rho family GTPases interact with the Arp2/3 complex to mediate actin polymerization and dynamic changes to actin structure, resulting in the formation of cortical-associated structures [5,6,7].

The actin-binding and nucleation-promoting factor cortactin (CTTN) is encoded by the *CTTN* gene (synonym: EMS1) located on chromosome 11q13 [2]. CTTN has several key structural domains, including an amino terminal acidic domain (NTA), a 6.5 tandem repeat domain consisting of a recurrent 37 amino acid sequence, a proline-rich region containing important phosphorylation sites, and the carboxy terminal Src-homology 3 (SH3) domain [2]. CTTN was originally identified as a substrate for Src kinase and plays key roles in the formation of cell membrane protrusions, such as lamellipodia and invadopodia, and in maintaining the integrity of the adherens junction cell-cell connections [8]. One major function of CTTN is activation of the Arp2/3 complex to promote branched actin polymerization at the cell periphery [2,9]. Cortactin shares high structural similarity with hematopoietic lineage cell-specific protein 1 (HS1), which is primarily expressed in hematopoietic cells [2,10]. Comparative genome analysis of CTTN and HS1 shows that both genes originated from a common gene duplication event [10]. They differ in some of their functions. For example, the F-actin binding domain of HS1 plays a role in apoptosis, while in cortactin it mediates cell migration [11]. The C-terminal SH3 domain of CTTN mediates protein-protein multiple interactions through which it provides an additional mechanism for regulating cellular processes. Table 1 lists some of the key proteins reported to interact with CTTN via its SH3 domain. These interactions can be induced by endogenous and other stimuli to produce cytoskeletal changes. As an example, hepatocyte growth factor (HGF) mediates cell proliferation and migration in a variety of cell types [12]. Our group previously reported that another endogenous factor, sphingosine-1-phosphate (S1P), promotes translocation of cortactin to the cell periphery and leads to the formation of a cortical ring to enhance barrier function in cultured human pulmonary artery endothelial cells (HPAECs), [13] and promote the interaction between the SH3 domain of CTTN and MLCK [14]. Figure 1 demonstrates that the interaction between CTTN and MLCK also occurs after HGF stimulation (as measured by the proximity ligation assay or Duo link assay).

### 1.2. Post Translational Modifications of Cortactin

CTTN is a major substrate for post translational modifications (PTMs), which are dynamic and reversible processes that regulate functional activities of proteins by the addition or deletion of functional groups [2]. The major PTMs reported for cortactin are ubiquitination, acetylation, phosphorylation, and glycosylation [47,48,49]. PTMs are critical for protein function and can be employed to generate properties relevant to therapeutic application [50]. As a result, one potential benefit of the improved understanding of the structure and functional relationships altered by key CTTN PTMs is that such knowledge may facilitate the downstream production of engineered proteins for possible therapeutic intervention for various disease processes.

#### 1.2.1. Phosphorylation and Dephosphorylation of Cortactin

CTTN contains several serine, threonine, and tyrosine phosphorylation sites close to the C-terminal end of the molecule that are important for integrating signals from multiple kinases and phosphatases to regulate key cellular mechanisms such as migration, permeability, inflammation, proliferation, and apoptosis [13,51,52]. Tyrosine phosphorylation occurs primarily on Y421, Y466 and Y482 sites (in the mouse protein; analogous sites in human CTTN are Y421, Y470, Y486) by Src family kinases, Abl kinases, FER and Syk [51]. Serine/threonine phosphorylation occurs primarily on S405, S418, T401 by extracellular signal-regulated kinases (ERKs), Pak1, and PDK [52]. CTTN phosphorylation promotes actin polymerization, tumor cell movement, and binding to FAK to activate cell scattering [53]. More specifically, CTTN tyrosine phosphorylation inhibits the activation of N-WASP, while serine phosphorylation of CTTN facilitates binding to N-WASP to stimulate actin polymerization [54,55]. Differential CTTN phosphorylation resulting in an opposite functional effect is known as the “S-Y switch model”. Activation of integrins α9β1 corresponds to the upregulation of the fibronectin matrix assembly and tyrosine phosphorylation of CTTN on Y470, which is correlated with adhesion strength and migration [56]. In addition, the cytoplasmic protein tyrosine kinase Syk associates with CTTN and promotes integrin-mediated phosphorylation on tyrosine, which correlates with the inhibition of cell motility in cancer cells [56,57]. Given its involvement in so many important cellular functions, a better understanding of the mechanisms underlying CTTN phosphorylation and downstream effects on cellular function is a promising area of investigation which may provide novel insights into processes common to several different disease-related processes [47,58].

#### 1.2.2. Ubiquitination and Deubiquitination of Cortactin

Ubiquitination is an important PTM which ensures cellular homeostasis. Ubiquitin-protein attaches to the substrate protein and controls protein degradation and membrane trafficking. CTTN degradation is also mediated by ubiquitination and regulates protein stability through the ubiquitin-proteasome system [52]. LPS-induced cortactin ubiquitination and degradation is regulated by ERK-serine phosphorylation in epithelial barrier regulation [52]. The role of CTTN ubiquitination was also explored in other pathways such as Wnt/β-catenin signaling and FBXL5 in cancer disease models. Hypermethylation of UCHL1 promotes cancer metastasis and downregulates CTTN degradation; however, it remains unclear how these CTTN ubiquitination PTMs relate to lung cell function or disease [59,60,61].

#### 1.2.3. Acetylation and Deacetylation of Cortactin

Another important CTTN PTM is acetylation regulated by the activities of histone deacetylase 6 (HDAC6) and sirtuin [62]. CTTN acetylation is implicated in the regulation of cell motility, spine morphogenesis and ciliary homeostasis, [63]. HDAC6 plays a role in microtubule-dependent cell migration by altering acetylation of CTTN and thereby changing its binding to F-actin binding [49]. Hyperacetylation of CTTN blocks the interaction with F-actin and impairs cell migration, which demonstrates the importance of HDAC6 in CTTN acetylation and cellular motility [63]. Keap1 is a cytosolic protein that plays an important role in oxidant stress responses and has been identified as a binding partner of CTTN. The interaction between CTTN and Keap1 is reduced by CTTN acetylation, which also regulates cellular migration, therefore suggesting a role of CTTN in modulating Keap1-associated oxidative stress responses [46]. The role of HDAC6-mediated CTTN deacetylation has been studied in ciliary dysfunction, in which overexpression of HDAC6 decreased acetylation of both tubulin and CTTN, and the overexpression of tubulin or CTTN enhanced the HDAC6-mediated ciliary disassembly [49,63]. These findings linking HDAC6 and actin polymerization in ciliary homeostasis provide mechanistic insight into the potential role of HDAC6 in actin dynamics that are altered in various disease processes [49]. There is also evidence that CTTN phosphorylation and acetylation are correlated in terms of functional effects since acetylation of CTTN decreases its phosphorylation at Y421 and reduces its interaction in neuronal cells with the adapter proteins p140Cap and Shank1, suggesting that the acetylation and phosphorylation of CTTN are inversely proportional to each other [63]. Many studies have identified that acetylation of CTTN lysine residues affects binding to F-actin, and that binding is further reduced when acetylated lysine residues are mutated to glutamine [49]. SIRT1 deacetylases CTTN, while p300 acetylates CTTN [64]. The knockdown of Sir2a, a mouse lung homolog of SIRT1, in mouse embryo fibroblasts resulted in decreased migration compared to control cells. In cancer cells, deacetylation of CTTN was increased when SIRT1 levels were high. Furthermore, the motility of cancer cells was reduced following the expression of an acetylated mimetic mutant of CTTN, while motility was increased when the cells expressed a deacetylation mimetic mutant [62,64].

#### 1.2.4. Glycosylation of CTTN

As noted above, co-translational and PTM alterations play key roles in regulating actin dynamics [65]. Glycosylation of proteins is an essential PTM in eukaryotic cells for protein folding and stability [66]. In human cancer cells, alterations in glycosylated proteins lead to the progression of cancer metastasis, a complicated process that results from multiple cytoskeletal-mediated events including migration, invasion, and epithelial-mesenchymal transition (EMT) [67]. Little is known regarding the potential role of CTTN glycosylation, and therefore further study of both the protein’s N-glycosylation and O-glycosylation may provide novel insights into various disease processes and could lead to the development of novel therapeutic interventions.

## 2. Vascular Endothelium and Epithelium

### 2.1. Vascular Endothelium

Endothelial cells play a critical role in maintaining vascular tissue homeostasis by forming the interface between the bloodstream and underlying vessel wall in large conduit vessels or organ-specific tissues surrounding capillaries [68]. This endothelial cell layer acts as a semipermeable barrier to fluid and solutes. Alterations to barrier function are hallmarks of many pathologic conditions including inflammation and edema. Endothelial dysfunction, characterized by increased permeability, plays an important pathologic role in several vascular processes mediated by CTTN (Figure 2: depicts role of CTTN in endothelial cell dysfunction). Endothelial cortactin depletion itself causes barrier dysfunction [69]. Additional studies focusing on the interplay between CTTN and endothelial dysfunction are discussed below.

Upon stimulation with various injurious stimuli, cytoskeleton rearrangement occurs, which leads to post translation modifications (PTMs) of CTTN. These PTMs can participate in pathophysiologic responses, including altered permeability, inflammation, invasion, migration and degradation mechanisms such as autophagy and apoptosis. Figure created using BioRender (available online https://app.biorender.com, accessed on 28 Ferbruary 2022).

#### 2.1.1. Cortactin and Other Cytoskeleton Proteins in Endothelial Barrier Regulation

Vascular barrier function results from a balance between contractile forces causing barrier disruption and tethering forces which regulate cell shape by the formation of a peripheral cortical ring linked to cell-cell and cell-matrix adhesions. CTTN plays an integral role in maintaining endothelial permeability by regulating cytoskeleton dynamics; for example, in response to sphingosine-1-phosphate (S1P), a barrier enhancer agent that promotes cortactin accumulation in the periphery and formation of lamellipodia, or thrombin, a barrier-disrupting agent that induces the opposite effects on CTTN and the peripheral actin structure [70,71]. The actin cytoskeleton plays a key role in regulating endothelial vascular permeability, and a study has shown that the elevation of Rho associated protein ROCK1 levels is associated with CTTN deficiency in regulating vascular permeability [69]. Also, ROCK1 inhibition leads to a reduction in actin-myosin contraction and restoration of vascular permeability caused by CTTN depletion, thereby suggesting counteracting roles of CTTN and ROCK1 in regulating vascular permeability [69]. HGF promotes endothelial cell motility and angiogenesis via the c-Met pathway. The rearrangement of CTTN and actin filaments is a driving force for lamellipodia formation in EC motility and promotion of the barrier function, and HGF enhances SphK1/p-SphK1 co-localization with actin/CTTN in this lamellipodia formation [72]. Another study reports that Asef-IQGAP1 and Arp3/CTTN interaction promotes HGF-induced endothelial barrier enhancement. This novel positive feedback mechanism promotes cytoskeleton rearrangement and regulates HGF-induced Rac activation [73,74]. The pharmaceutical compound simvastatin also promotes barrier-enhancing cytoskeletal rearrangements via Rac-dependent signaling in association with CTTN translocation to the EC periphery. Simvastatin attenuates thrombin-induced endothelial barrier dysfunction and MLC phosphorylation. It also reduces stress fiber formation in association with CTTN translocation, thereby leading to reduced gap formation [75].

Myosin light chain (MLC) is phosphorylated by the Ca^+2^/calmodulin-dependent enzyme myosin light chain kinase (MLCK) as a critical step in generating actomyosin interaction and cell contraction [14,76]. MLC phosphorylation and subsequent force generation is associated with actin cytoskeletal rearrangement [13,76,77,78]. CTTN and MLCK interact at baseline in endothelial cells (EC), but this interaction is reduced after stimulation with the barrier-disruptive agent thrombin [79]. In contrast, S1P stimulation increases the CTTN-MLCK interaction. The role of the CTTN SH3 domain is critical for this interaction, as CTTN lacking the SH3 domain showed significantly decreased interaction with MLCK after S1P challenge. The subcellular location of CTTN and MLCK interaction plays an important role in determining the effects on vascular barrier regulation [80]. Abl kinases phosphorylate CTTN, and biomechanical studies investigating the role of Abl kinases on the endothelial cytoskeleton structure suggest a form-function relationship between Abl and CTTN. For example, after stimulation by the barrier-promoting endogenous compound S1P, barrier enhancement occurs due to actin cytoskeleton rearrangement. However, the redistribution of actin and its association with CTTN, as well as barrier enhancement, are reduced when c-Abl kinases are inhibited by imatinib, suggesting a potential role of Abl kinases in mediating S1P-induced actin-CTTN association [81]. In a biomechanical study using atomic force microscopy to measure elastic modulus properties in lung endothelium, reductions in c-Abl kinase expression by siRNA decreased phosphorylation of MLCK and CTTN and reduced the EC elastic modulus, resulting in the attenuation of barrier enhancement [76,81].

#### 2.1.2. Cortactin in Cellular Migration

S1P also induces capillary formation in EC, facilitating migration due in part to an increased association of CTTN with the Arp2/3 complex [24]. Attenuated CTTN expression via CTTN antisense oligo impairs S1P-induced endothelial migration; however, overexpression of cortactin altered interaction with Arp2/3 complex, providing further support that interaction between the Arp2/3 complex and CTTN plays a vital role in S1P-mediated actin polymerization and the remodeling of EC [24]. Phospholipase D (PLD) produces phosphatidic acid (PA) and participates in another signaling response that induces cell migration. PA binds to the tyrosine kinase Fer and enhances phosphorylation of CTTN to promote cell migration via actin polymerization [82].

#### 2.1.3. Cortactin and Reactive Oxygen Species

Injurious levels of reactive oxygen species (ROS) adversely affect intracellular functions resulting in inflammation and potentially cell death, and ROS also can reduce mitochondrial membrane potential to induce mitochondrial dysfunction and contribute to various lung diseases [83]. CTTN regulates NADPH oxidase activation and ROS formation in association with Src kinase-dependent tyrosine phosphorylation of p47^phox^ [84]. Hyperoxia increases tyrosine phosphorylation of CTTN and its interaction with p47phox in human pulmonary artery EC, while hyperoxia-induced generation of ROS is significantly lowered in the tyrosine-deficient mutant of CTTN when compared to the wild type [78,84].

#### 2.1.4. Cortactin in Endothelial Apoptosis

Exposure to cigarette smoke (CS) induces apoptosis in lung endothelial cells and is a key feature contributing to the pathophysiology of COPD [58]. We recently identified a role for CTTN in CS-induced lung endothelial lung injury, demonstrating downregulation of CTTN mRNA levels in smokers compared to nonsmokers, cytoskeletal rearrangement causing an increase in actin stress fibers in lung EC, and increased CTTN phosphorylation upon CS challenge. Lung endothelial apoptosis and mitochondrial ROS levels were elevated upon CS challenge. Interestingly, these effects were accentuated in cells deficient in CTTN (via siRNA transfection), thereby highlighting a novel role for CTTN in CS-induced lung endothelial dysfunction and possibly COPD pathology [58].

### 2.2. Airway Epithelium

Respiratory epithelial cells line the airway from the trachea to the bronchi and into the distal bronchioles and alveolar sacs [85]. This airway epithelium is the first line of defense against inhaled lung pathogens and plays a critical role in multiple lung responses. Compared to vascular EC, the role of CTTN in the airway epithelium remains less well-defined.

#### Cortactin in the Airway

Autotoxin (ATX) is an endogenous stimulatory factor expressed in lung epithelial cells that promotes cancer cell motility and tumor metastasis in lung cancer. The novel role of ATX and PKCδ-mediated cortactin phosphorylation was explored in relation to migrating epithelial cells. This study identified a role of ATX in cell migration through the activation of LPA receptors. This activation leads to cytoskeletal rearrangement, phosphorylation of PKCδ and CTTN in migrating cells, which may contribute to airway re-epithelialization and remodeling after injury [86]. The interaction between CTTN and the cytoskeletal protein XB130 provides further evidence for a role for CTTN in epithelial cell migration. CTTN-XB130 interaction was enhanced by treatment with nicotine derived nitrosamine ketone (NNK), a nicotine-derived cigarette smoke component produced in the airway epithelium. Furthermore, NNK induced peripheral cytoskeletal rearrangement and the association between F-actin and CTTN, processes which promote actin polymerization and cell migration [87].

Lipopolysaccharide (LPS) induces serine phosphorylation of CTTN by ERK and causes its degradation and ubiquitination in the lung epithelium [52]. ERK-mediated serine phosphorylation of CTTN is crucial for CTTN ubiquitination and degradation upon LPS challenge. Mutation of these serine phosphorylation sites in CTTN (S405A/S418A) results in CTTN protein stability. The E3 ligase subunit β-Trcp interacts with CTTN, and its overexpression reduces CTTN protein levels, suggesting that CTTN stability is regulated by the ubiquitin-proteosome network [52]. Therefore, cortactin stability is regulated by ERK and β-Trcp controlling epithelial barrier function. Influenza A virus (IAV) infection of epithelial cells results in degradation of CTTN, with CTTN undergoing ubiquitination during IAV infection in a lysosome-associated apoptotic pathway. RNA interference and overexpression methods revealed an association between CTTN degradation and IAV infection in the epithelium. This study also suggested the presence of multiple caspase cleavage sites on the CTTN polypeptide [88].

## 3. Acute Lung Injury

### 3.1. Pathophysiology of Acute Respiratory Distress Syndrome and Acute Lung Injury

Acute Respiratory Distress Syndrome (ARDS) is an inflammatory lung condition characterized by disruption of endothelial barrier integrity, increased vascular permeability, and resulting alveolar edema leading to a loss of gas exchange between the lung alveolar space and the pulmonary circulation. ARDS is a leading cause of morbidity and mortality in critically ill patients, with ~200,000 estimated cases each year in the US prior to the COVID-19 pandemic [89,90]. ARDS causes “diffuse alveolar damage” (DAD) as a histopathologic correlation, which is characterized by alveolar epithelial cell necrosis, inflammatory cell infiltration, and excessive accumulation of neutrophils, hyaline membrane formation within the alveoli, interstitial edema, and alveolar-capillary disruption. The most common causes of severe ARDS are pneumonia and sepsis, but there is a lack of knowledge on effective and specific treatment for the underlying pathophysiology of this condition (Figure 3) [89,90]. Despite recent advances, the pathophysiology of ARDS remains incompletely characterized. Multiple studies have been performed utilizing preclinical models to better understand the mechanisms underlying the acute lung injury (ALI) process that results in clinical ARDS. An improved mechanistic understanding is needed to assist in the development of targeted therapies. Examples of pathways explored in preclinical models of ALI include S1P receptors and S1P metabolizing enzymes, Abl kinase signaling and imatinib inhibition, and CTTN-MLCK interactions [13,91,92,93,94].

This figure illustrates some functional changes that have been reported to involve CTTN and can be associated with important pulmonary disease processes. For example, ronchoconstriction is associated with asthma, aspirin exacerbated respiratory disease (AERD), and alveolar disruption with increased permeability and mucous production is associated with ALI/ARDS, bronchitis and emphysema with COPD, and broncho pulmonary dysplasia (BPD). Tumor formation and invasiveness is associated with lung cancer. Figure created using BioRender (available online at https://app.biorender.com, accessed on 28 Ferbruary 2022).

### 3.2. Role of Actin Cytoskeletal Rearrangement in Lung Endothelial Barrier Regulation

As above, a key feature of ALI is loss of the barrier between the vascular and alveolar spaces in a process that involves dysfunctional epithelial and endothelial cell layers. The next section briefly provides several examples of the importance of cytoskeletal structures in ALI responses. CTTN is likely to play a role in some of these responses.

### 3.3. Role of Prostaglandin A2-EP4 in ALI

Prostaglandin PGA2 demonstrates protective effects in an ALI model induced by LPS. PGA2 suppressed LPS-induced inflammatory responses by inhibiting the NFkB pathway and reducing the expression of endothelial adhesion proteins ICAM1 and VCAM1. PGA2 mediates these effects by inhibiting EP4, which suggests that the PGA2-EP4 pathway has potential for vascular endothelial protection and for ameliorating the vascular leakage and inflammation that underlies ALI [95].

### 3.4. Role of Asef in Cytoskeleton Remodeling in HGF-Mediated ALI

Asef is a guanine nucleotide exchange factor (GEF) that alters cytoskeletal dynamics in ALI. An Asef mutant leads to HGF-induced peripheral actin cytoskeleton rearrangement and the enhancement of Rac1 activity, while knockdown of Asef expression causes a decrease in HGF-induced Rac1 activation. Asef knockout causes enhanced lung inflammation and vascular leak, suggesting a role for Asef in Rac activation and protection against endothelial permeability and vascular injury [96].

### 3.5. Polyethylene Glycol in Actin Cytoskeleton Rearrangement

Polyethylene glycol mimics the mucin lining in the epithelium and attenuates both thrombin- and LPS-induced endothelial dysfunction. It activates a barrier-enhancing pathway targeting actin cytoskeleton to ameliorate inflammatory lung injury, suggesting that related agents may be potential ALI therapies [97]

### 3.6. Cytoskeletal Activation and Altered Gene Expression Regulated by Simvastatin

The pharmaceutical compound simvastatin affects endothelial barrier function, actin myosin contraction, and gap formation. In addition to increasing the activity of Rac GTPase activity, simvastatin promotes actin-CTTN interaction and attenuates thrombin-induced MLC phosphorylation. These cytoskeleton changes combine with gene expression changes in response to simvastatin to exert positive effects on ALI-related processes [75].

### 3.7. FTY720-S1P1 Receptor in Barrier Enhancement

FTY720 is a pharmaceutical agent that is also an analog of S1P. Similar to S1P, FTY720 exhibits barrier protection in lung EC and some preclinical models of ALI, but aspects of its mechanistic effects are different. While S1P induces significant MLC phosphorylation and actin cytoskeleton rearrangement in lung EC, FTY720 does not, and reduced expression of Rac1 or CTTN attenuates S1P-induced barrier enhancement, but not that induced by FTY720 [13,98,99,100].

### 3.8. Cortactin Genetic Variant in ARDS

The risk and severity of complex disease processes such as ARDS are associated with gene variation, which highlights the need for characterizing the role of genetics in ALI [101]. Genome-wide association studies and microarray analyses have been performed in ARDS patients to identify disease-related single nucleotide polymorphisms (SNPs). We have identified a human disease-related CTTN variant in which an SNP results in serine at position 484 being mutated to asparagine (S484N) [101]. This SNP alters CTTN tyrosine phosphorylation at the nearby key Y486 site and inhibits endothelial barrier function, cell migration speed and directionality, alters lamellipodia dynamics, and impedes wound healing [102]. These in vitro observations in the cultured lung endothelium represent processes relevant to ALI pathophysiology. Recent confirmatory data have demonstrated that this CTTN S484N SNP increases ALI susceptibility in mice and is associated with increased mortality in patients with severe sepsis, as well as increased risk and severity for acute chest syndrome (a form of ALI) in sickle cell patients [103]. These studies support an important role for CTTN in ALI pathophysiology.

## 4. Chronic Obstructive Pulmonary Disease (COPD)

### 4.1. Pathophysiology of COPD

Chronic obstructive pulmonary disease (COPD) is primarily caused by cigarette smoke (CS) and is a leading cause of mortality [52,104,105]. Despite recent advances, there is no effective treatment available to reverse smoking-related lung damage. COPD is an inflammatory disease with increased macrophage and neutrophil infiltration which leads to narrowing of the airways, airflow limitation, and destruction of alveoli (Figure 3). Other key features of COPD include remodeling of the airways and loss of elastic properties of the lung parenchyma due to the release of elastases, chemokines, and cytokines by the immune cells [104]. The risk factors associated with the development and progression of COPD are tobacco smoking, environmental exposures, infections, and other genetic risk factors [105]. Continued exposure to nicotine and other toxic components of CS leads to the development of chronic bronchitis [106,107].

### 4.2. CS-Induced Responses in COPD

Multiple studies have investigated the mechanisms of COPD and explored the development of novel therapies for CS-induced COPD. The basic pathogenesis of CS-induced COPD involves airway inflammation, which results in functional and structural changes in the airways. Exposure of the lung to CS increases production of ROS and affects many other molecular mechanisms [58,108,109,110]. Upon CS challenge, autophagy is induced, thereby increasing the SESN2, ATG12, and LC3B levels. NAC (N-acetyl cysteine) supplementation reduced ROS levels, and these molecular changes were reversed, suggesting the role of ROS, SESN2, and mTOR in the CS-induced autophagic process. The PI3K/AKT/mTOR pathway was also studied in CS-induced actin cytoskeleton rearrangement, cellular senescence, and accelerated lung aging and inflammation [12]. CS-induced ROS production affects the morphology and function of both lung epithelium and endothelium and disrupts adherens junctions, decreases the activity of Nrf2, and reduces E-cadherin expression levels [104].

### 4.3. Role of CTTN in Cigarette Smoke-Induced Cell Migration and Invasion

The mechanisms underlying cell migration and invasion are important to better characterize the progression of various disease processes [111]. As above, Arp 2/3 and CTTN are key actin cytoskeletal proteins involved in cell migration and invasion [2,3]. Altered cell migration driven by changes in actin polymerization may play a role in the pathogenesis of several lung disorders. CS exposure in human endothelial cells leads to actin cytoskeletal changes [58]. The primary changes associated with exposure to CS include the induction of oxidative stress and an increase in intracellular calcium concentration. CS-induced actin cytoskeleton rearrangement such as F-actin assembly was attenuated when treated with antioxidants and PLC-IP3-PKC, suggesting an important role for PLC-IP3-PKC in these actin rearrangements [112]. The effects of commercially available CS condensate on migration and survival suggest that lower concentrations may stimulate cellular migration and invasion, but higher concentrations attenuate the effect [113].

A two-hit model in which bronchial epithelial cells were stimulated with CS and LPS increased actin polymerization and altered the expression of proteasome activator complex subunit 2 (PSME2), peroxiredoxin-6 (PRDX6), annexin A5 (ANXA5) heat shock protein beta-1 (HSPB1) and Coactosin-like protein (COTL-1). This study revealed the role of cytoskeleton proteins involved in actin cytoskeleton dynamics upon LPS and CS induced lung injury [114]. As above, Rho-GTPase are key regulators of actin cytoskeleton organization, and a genome-wide expression study on CS-exposed lung embryos suggested the upregulation of Rho-GTPase [115], suggesting a role for these signaling molecules in CS responses.

Cluster of Differentiation-44 (CD44) is a glycoprotein that plays a key role in multiple mammalian cell biological functions, and overexpressed CD44 is found in some tumors such as lung cancer [116].The potential role of CD44 receptor biology in CS-induced lung injury mechanisms remains largely unknown. Ouhtit et al. explored the role of CD44 and CTTN/survivin to investigate the signaling pathways during lung injury, which may have potential to serve as a therapeutical target [117]. Cigarette smoking disrupts mesenchymal stem cell repair mechanisms, thereby potentially contributing to the development of lung cancer. Glycosaminoglycans (GAGs), such as hyaluronic acid (HA), have been explored as a possible therapy for lung injury [117]. HA/CD44 are transcriptional targets for CTTN and survivin and CS attenuated CD44 and CTTN expression in MSCs [117]. Furthermore, silencing CD44 inhibited migration and invasion of MSCs upon CS challenge. This suggests the role of CD44/CTTN signaling mechanism in CS-induced respiratory lung injury and identifies a new therapeutic target [117].

### 4.4. CTTN Methylation in COPD

DNA methylation is a common epigenetic signaling tool in eukaryotes that controls gene expression [118]. DNA methylation is important for various cellular processes, and abnormal DNA methylation has been associated with human disease, but the mechanisms involved remain incompletely defined. CpG islands are regions for DNA methylation present in promoters known to regulate gene expression. A recent epigenetic study characterized the association of DNA methylation with cigarette smoking [119]. A genome-wide DNA methylation meta-analysis of 15,907 blood-derived DNA samples revealed 1405 differentially annotated genes in smokers compared to controls. These data suggest that these genes annotated to CpGs may have an association with CS-induced pulmonary dysfunction. The goal of this analysis was to identify possible novel molecular mechanisms and pathways associated with CS-induced diseases. This analysis identified the CTTN gene as exhibiting a significant increase in DNA methylation among current versus never smokers. Furthermore, this study suggests the potential role of CS-induced DNA methylated genes in various vital molecular processes and the need to address its effects on human health and disease [119].

## 5. Asthma and Other Lung Disorders

### 5.1. Pathophysiology of Asthma and CTTN Gene Variation as a Key Regulator of Cytoskeleton Rearrangements

Asthma is a chronic inflammatory disorder of the airways. According to the Centers for Disease Control and Prevention, 4.3% of the world population i.e., ~300 million suffer with asthma [120]. Key pathophysiologic features of asthma include bronchoconstriction- narrowing of airways, airway edema, airway hyperresponsiveness and airway remodeling (Figure 3) [120,121]. Despite decades of research, the pathophysiology of asthma remains incompletely understood. Major asthma triggers including respiratory infections, airborne allergens, cold air, pollutants, and irritants and differ from person to person. Genetic factors play a significant role in asthma phenotypes, and better understanding of asthma-associated genes and their interactions are an important area of current study. Genome-wide association studies have been used to identify over 100 gene mutations and polymorphisms with a possible role in asthma pathophysiology, including associations with the regulation of chemokines, myriad cytokines, and growth factors that may influence the development and progression of disease [122].

Candidate gene association studies identified chromosome 11q13 as being linked with five asthma related genes (FCER1B, CC16, GSTP1, GPR44 and IL18) [123,124]. MYLK, the gene encoding the cytoskeletal force generating contractile protein MLCK (see above), is strongly associated with severe asthma, and there is an association between MYLK and CTTN [14]. The CTTN gene was identified to be in proximity with asthma-related genes in the 11q13 chromosome. A total of 9 CTTN gene polymorphisms (SNPs) associated with asthma were identified, and a specific intronic SNP, rs3802780, showed a significant association with severe asthma, suggesting the importance of CTTN in altering the clinical phenotype [123].

### 5.2. CTTN PTMs in Pathogenesis of Asthma

#### 5.2.1. Cortactin in Association with Profilin in Pathogenesis of Asthma

Profilin is a small actin-binding protein which plays an important role in actin polymerization. In higher concentrations, profilin inhibits actin polymerization, while at lower levels it enhances actin polymerization [125,126]. Profilin alters cytoskeletal structure through Rho-GTPase binding, as well as binding to CTTN through a proline rich region to regulate cytoskeleton rearrangement, actin polymerization, and cell migration [77,125,126]. Profilin associates with CTTN in smooth muscle contraction. Changes in smooth muscle cell contraction led to reduced airways resistance, which contributes to the pathogenesis of asthma. Other groups have reported the role of profilin in actin polymerization and migration/invasion. Actin smooth muscle contraction requires MLC phosphorylation at Ser-19 and actin reorganization [125]. Previous work implicates CTTN phosphorylation at Y421 in the regulation of cell migration and c-Abl through regulation of actin dynamics, proliferation, and cell adhesion [126,127,128]. Recent studies showing activation of c-Abl due to smooth muscle contraction lead to the investigation the role of profilin interaction with CTTN and c-Abl. The c-abl tyrosine kinase phosphorylates CTTN at 421 site, and the phosphorylated CTTN facilitates the accumulation of profilin at cell edges promoting actin polymerization and cell movement [126,127]. Thus, profilin plays a vital role in the regulation of airway hyperresponsiveness and smooth muscle contraction via processes involving CTTN and c-Abl, suggesting a potentially novel role of CTTN in asthma pathogenesis [125].

#### 5.2.2. Cortactin in Aspirin-Exacerbated Respiratory Disease

Aspirin-exacerbated respiratory syndrome (AERS) is a chronic inflammatory condition characterized by the combination of asthma, chronic rhinosinusitis, and sensitivity to aspirin (Figure 3). The participation of CTTN in inflammatory responses, airway secretion and contractile mechanisms have been previously reported [125,129]. CTTN expression was assessed in nasal polyps from AERS patients by immunohistochemistry [125]. High levels of CTTN expression were found in both the epithelium and fibroblasts of patients with AERS when compared to the control group. The expression of CTTN was also reported to be higher in nasal polyps of females when compared to males, suggesting a possible explanation for differences in AERS severity. However, further investigation of gender differences in CTTN-mediated AERS are needed [129].

#### 5.2.3. Deacetylation of Cortactin

HDAC8 is a class 1 histone deacetylase enzyme present which regulates nonhistone protein deacetylation and plays a significant role in DNA repair, as well as maintaining and organizing chromosomes during cell division. HDAC8 also plays a key role in smooth muscle contraction [130]. A recent study reported that acetylcholine induces CTTN deacetylation in smooth muscle. Furthermore, HDAC8 and CTTN both mediate activation of smooth muscle contraction [130]. Actin polymerization increases after CTTN deacetylation by SIRT1 and HDAC6 [49]; however, in the present study it was reported that HDAC8 knockdown or inhibition attenuated actin polymerization, suggesting a potential role of HDAC8 in contractile activation [130]. Interestingly, MLCK does not appear to play a role in HDAC8-mediated actin polymerization. Further, a 9KQ mutant CTTN (dominant-negative mutant mimicking cortactin acetylation) altered smooth muscle contraction and actin polymerization. This study established a novel mechanistic role for CTTN in HDAC8-mediated actin polymerization [130].

#### 5.2.4. Shear Stress in Cortactin and Actin Polymerization

Mucus hypersecretion is accompanied by accentuated shear stress due to bronchoconstriction in obstructive airway diseases such as COPD and asthma. Previous work has reported a role for the mucus-related protein mucin 5AC in actin polymerization. The role of MUC5AC has also been explored in actin polymerization via tyrosine phosphorylation of CTTN in airway epithelial cells. In these cells, increased shear stress was associated with elevated Src phosphorylation, CTTN phosphorylation, actin polymerization and MUC5AC secretion in CTTN-overexpressing cells, while all these effects were attenuated in cells receiving siRNA to reduce cortactin expression [58,131]. While CTTN has been extensively studied in the endothelium, this study demonstrates a novel role of shear stress induced MUC5AC secretion via CTTN actin polymerization in airway epithelial cells [71,131].

### 5.3. Cortactin in Broncho Pulmonary Dysplasia

Bronchopulmonary dysplasia (BPD) is a chronic lung injury resulting from premature birth that leads to reduced pulmonary function [132]. It is characterized by decreased cellular migration and proliferation (Figure 3). Studies have reported the role of actin binding proteins, cofilin, profilin, and VASP in regulating proliferation and migration via actin rearrangements. Variations in the expression of profilin 1 and cofilin 1, and phosphorylation of VASP, suggest potential roles for these actin-binding proteins in BPD pathogenesis [132]. While a specific role for CTTN in BPD pathogenesis is unknown, it associates closely with these other mediators of cytoskeletal rearrangement in similar cell processes [133,134], and thus it is plausible to speculate a role for CTTN in BPD pathology.

### 5.4. Cortactin in Lung Cancer

An extensive body of literature has explored the association of CTTN structure and function with various types of malignancies [135,136,137], with some recent reviews published elsewhere [8,135]. A detailed discussion of CTTN in cancer is beyond the scope of this current review, but Table 2 lists multiple proteins and other regulatory molecules reported to interact with CTTN and mediate oncologic processes. Here we highlight a few recent topics related to CTTN expression and function in lung cancer. For example, a recent report suggests that the chemokine CX3CL1 promotes lung cancer invasion and migration by elevating the levels of c-Src and c-Abl to increase CTTN phosphorylation, since inhibition of src/abl, or mutating the phosphorylation sites in CTTN, blocks lung cancer invasion and migration [138]. Another study correlated immunohistochemistry and clinical data to demonstrate that CTTN (and SIRT1) expression is significantly increased in non-small cell lung cancers (NSCLC), and it is associated with high pathological lymph node metastasis, tumor invasion, and shorter survival, which suggests a potential role for CTTN in NSCLC progression [62]. A different NSCLC study describes the role of CTTN and its binding partner dynamin in stimulating the formation of F-actin bundles, leading to filopodia formation and cancer cell migration [139]. Many studies have reported microRNA involvement in regulating cancer metastasis. Modulation of CTTN by microRNAs has been reported in association with several types of cancer, including in lung cancer via mi-R-182 and miR-509 [136,140,141]. To briefly summarize, these reports and others suggest an important role for CTTN in regulating cell migration and invasion in lung cancer, and therefore targeting cortactin may be a potential anticancer therapy worth further exploration.

### 5.5. Role of Host Cytoskeleton in Coronavirus Infection

Severe acute respiratory syndrome coronavirus-2 (SARS-CoV-2) causes a highly infectious respiratory disease that is responsible for the current global pandemic [153]. After viral invasion into the host airway epithelial cells, viral particles disrupt the host cytoskeleton and interrupt the assembly of actin binding proteins [154]. Viral infection is exacerbated by altering the host cell homeostatic mechanisms and affecting cytoskeleton regulatory signaling pathways [155,156]. Viruses manipulate the actin cytoskeleton of the host to promote cell entry and the propagation of virus. Actin dynamics are implicated in SARS-CoV influence on the MAPK cascade [157]. Previous work on coronavirus shows that Rho family Rac1/Cdc42 small GTPases were used by coronavirus-transmissible gastroenteritis and porcine hemagglutinating encephalomyelitis virus to promote actin polymerization and reorganization [156,157,158]. The role of dynamin, a microtubule related protein, which interacts with the CTTN SH3 domain, was studied in the SARS CoV-2 entry process into the host cell [159]. Therefore, cytoskeletal regulation is a rich area for exploration given its relationship to virus entry, and further study of CTTN and related signaling mechanisms may help identify key mechanistic targets for the development of novel therapies.

## 6. Conclusions and Future Directions

Vascular endothelial dysfunction plays an important role in multiple lung diseases. Despite recent advances, the underlying mechanisms of CTTN in various lung injury models remain incompletely defined. Several promising areas of additional research are likely to provide further insights into the functional roles of CTTN in lung health and disease, including CS effects on mitochondrial function, apoptosis, autophagy, and permeability. Further investigation will better define the role of CTTN in vascular barrier function, inflammation, and airway smooth muscle in the pathophysiology of asthma and COPD. Some key post-translational CTTN modifications are still poorly understood, such as nitration and glycosylation, and may provide novel therapeutic targets for various inflammatory lung diseases such as ARDS. In summary, further study of the many pathophysiologic mechanisms involving CTTN holds promise for advancing our understanding of multiple problematic lung diseases.

## Figures and Tables

**Figure 1 ijms-23-04606-f001:**
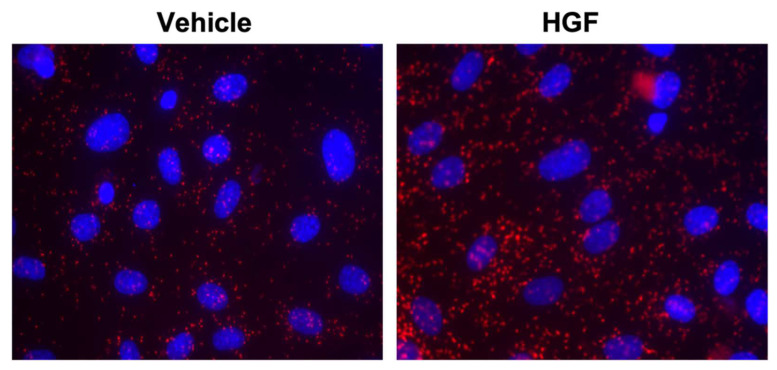
In vitro analyses showing interaction of cortactin with MLCK in lung endothelial cells. Serum starved HPAECs were incubated with either control vehicle or HGF for 5 min, and then the association of MLCK and CTTN was detected using in situ proximity ligation assay (red dots). Representative images are shown.

**Figure 2 ijms-23-04606-f002:**
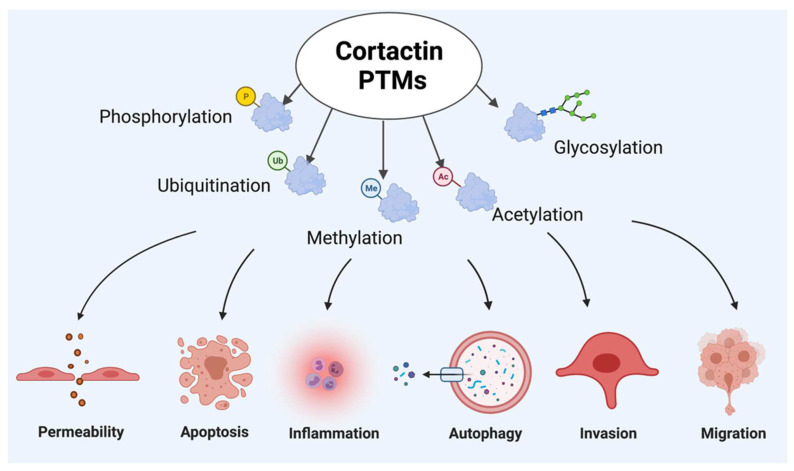
Cortactin post-translational modifications (PTMs) participating in endothelial cell pathophysiologic responses.

**Figure 3 ijms-23-04606-f003:**
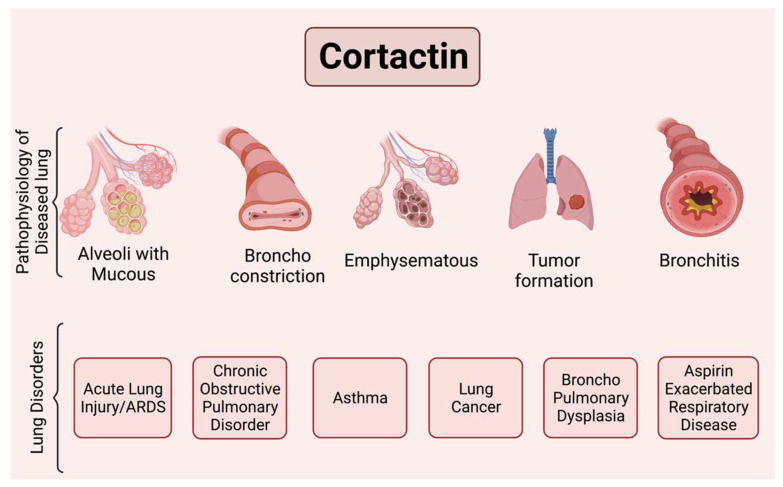
Scheme illustrating major lung pathologies mediated by cortactin.

**Table 1 ijms-23-04606-t001:** Representative key proteins that interact with CTTN.

	Protein	Site of Interaction with CTTN	References
1	SHANK2 and SHANK3	SH3 domain	[15]
2	FGD1	SH3 domain	[16]
3	ABL2	Homology	[17]
4	KCNA2	C terminus	[18]
5	SAMSN1	SH3 domain	[19]
6	ASAP1	SH3 domain/Pro-rich region	[20]
7	DNM2	SH3 domain/homology	[21]
8	ACTN1	Homology	[22]
9	N-WASP	SH3 domain	[23]
10	Arp2/3 complex	NTA/SH3 domain	[24]
11	Kv1.3/Kv1.2	CTTN	[18,25]
12	Caspase cleavage sites	Actin binding domains/SH3 domain	[26,27]
13	UCS15A/AMAP1	SH3 domain/pro-rich region	[28,29]
14	STAM1, TXK, Fyn, Hck	SH3 domain	[30]
15	AFAP1L1	SH3 domain	[31]
16	Calcium activated potassium channels	SH3 domain	[32]
17	CD2AP	SH3 domain	[33,34]
18	TIP150	C terminal tail	[35]
19	CBP90	SH3 domain	[36]
20	Cytoplasmic tyrosine kinase FER	CTTN	[37]
21	MLCK	SH3 domain	[14]
22	Cdc42-associated kinase 1(ACK1)	SH3 domain	[38]
23	FAK	SH3 domain	[39]
24	Protein ZO-1	SH3 domain	[40]
25	Lymphocyte protein 2(Sly2)	SH3 domain	[41]
26	WIP	SH3 domain	[42]
27	Fgd1, Cdc42, GEF	SH3 domain	[43,44]
28	BPGAP1	SH3 domain	[45]
29	Keap1	CTTN	[46]

**Table 2 ijms-23-04606-t002:** Role of CTTN in association with other signaling components in regulating cellular mechanisms in lung cancer (Figure 3).

	Protein	Function	References
1	miR-182	↓ invadopodia formation and metastasis Inhibits invasion and proliferation	[136,142,143]
2	Isoliquiritigenin,2,4,2′,4, Tetrahydroxychalcone metabolite	↓ lung cancer invasion	[137]
3	SIRT1 (sirtuin1)	Involved in tumor progression	[62]
4	Dynamin1	Cell migration, stabilizes filopodia formation	[140,141]
5	Actinin-1, Ect2	Localization of invadopodia, matrix degradation and migration	[22]
6	MT1-MMP, Tks4, Tks5	Invadopodia during cancer extravasation and metastasis	[142]
7	Protein Kinase D1	↑ extracellular vesicles, promotes metastasis	[143]
8	Vascular endothelial growth factor- C	Promotes metastasis	[144]
9	CAIX-Tumor associated carbonic anhydrase IX	Promotes Invadopodia formation and matrix degradation	[145]
10	Dynamin2	Suppresses lamellipodia formation and invasion	[139]
11	P140Cap	Suppresses invasion of MTLn3-EGFR	[146]
12	G-protein-coupled receptor-2-interacting protein-1	Directional migration and tumor angiogenesis, lamellipodia formation	[147]
13	XB130	Mediates NNK-induced migration	[87]
14	Class II HDAC6	Endothelial cell migration and angiogenesis	[148]
15	Abl Kinases	Cancer progression	[128]
16	ZMAT3	Knockdown of CTTN/ZMAT3 ↓ cell survival	[149]
17	EphA2 mutation	Promotes cell survival, cell invasion and mammalian target of rapamycin activation	[150]
18	CD44	HGF mediated vascular integrity	[116,151,152]
19	CX (3) CL1	Cell invasion and migration	[138]

## Data Availability

The data presented in this study is available upon request from the corresponding author.

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
