# Peer review of "Cortactin in Lung Cell Function and Disease"

_ijms, 2022, doi:10.3390/ijms23094606_

Round 1
Reviewer 1 Report
In this review the authors summarize the evidence for the cellular function of cortactin. As this protein binds actin it exerts an important cytoskeletal role with respect the membrane integrity and cell migration both in lung cell physiology and pathophysiology. This review touches on important aspects in the field, and should be published in IJMS after minor revisions.
Figure 2 and 3 legend: I would replace the word graphic by figure.
Figure 3 legend: there are a few words that should not be capitalized – eg.: bronchitis, emphysema, etc.
Are there any data on cortactin in patients that would corroborate the in-vitro findings (eg.: liquid biopsy samples or studies in FFPE lung cancer tissue)?
Section 3: Lung injury - similar to above: are there any clinical data on the role of this protein in patients who experienced ARDS?
Table 2 mentions micro-RNA-182 as a synergistic factor for CTTN with respect to cell invasion and proliferation – are there additional microRNAs that co-regulate CTTN on an epigenetic level (or other ncRNAs)? Are there any other epigenetic factors that play a role in the regulation of CTTN (eg.: the methylome)?
Although I know that it is not the main focus of the article and the authors state that it is outside the scope of this review, I would suggest to shed light on CTTN in the formation of lung cancer. As a researcher in the field of thoracic cancer I am curious about the role of CTTN, therefore I would expand section 5.4.

Author Response
Reviewer #1: In this review the authors summarize the evidence for the cellular function of cortactin. As this protein binds actin it exerts an important cytoskeletal role with respect the membrane integrity and cell migration both in lung cell physiology and pathophysiology. This review touches on important aspects in the field and should be published in IJMS after minor revisions.
- Figure 2 and 3 legend: I would replace the word graphic by figure.
Thank you for this suggestion. These changes have been made in the legends for Figure 2 and 3.
- Figure 3 legend: there are a few words that should not be capitalized – eg.: bronchitis, emphysema, etc.
We apologize for these errors. We have reviewed and corrected them in the figure legend.
- Are there any data on cortactin in patients that would corroborate the in-vitro findings (eg.: liquid biopsy samples or studies in FFPE lung cancer tissue)?
We certainly agree that patient-related data are valuable and important for confirmation of in vitro observations. There are multiple reports of cortactin overexpression in tissue samples from various types of malignancies (e.g., squamous cell cancer of head and neck, breast cancer, etc.) that correlate with increased tumor invasiveness and/or poorer clinical outcomes. Since the extensive cancer literature is not the focus of our review (although in this revision we have added discussion of recent lung cancer work—see responses to #5 and #6 below), we have not described these reports in detail. However, in our own recently published work describing a functional role for CTTN in lung EC responses to cigarette smoke exposure (PMID: 34831092), we evaluated a small set of lung tissues samples from chronic cigarette smokers and extracted RNA to perform qPCR. These data suggest that CTTN mRNA levels are lower in smokers compared to non-smokers, which is supportive of a potential role in smoking-induced pathophysiology. These data are briefly mentioned in this review in section 2.1.4.
- Section 3: Lung injury - similar to above: are there any clinical data on the role of this protein in patients who experienced ARDS?
We also agree with the reviewer regarding the importance of clinical data to confirm the possible role of CTTN in lung injury models. To that end, we also now have published a study describing functional effects of a single nucleotide polymorphism (SNP) in the CTTN gene that is associated with worse clinical outcomes in two different patient populations. These data are briefly described in section 3.8.
- Table 2 mentions micro-RNA-182 as a synergistic factor for CTTN with respect to cell invasion and proliferation – are there additional microRNAs that co-regulate CTTN on an epigenetic level (or other ncRNAs)? Are there any other epigenetic factors that play a role in the regulation of CTTN (eg.: the methylome)?
This is another important point raised by the reviewer. There is a substantial amount of literature indicating that microRNAs play important roles in cancer invasion and metastasis in general, as well as several studies relevant to CTTN expression more specifically. Modulation of CTTN by microRNAs has been reported in association with several types of cancer, including in lung cancer via mi-R-182 and miR-509. Some representative studies are now described in the expanded and completely revised lung cancer section 5.4. Regarding epigenetic regulation, one very intriguing study demonstrated that methylation in the CTTN gene is one of the most highly altered genes in the lungs of smokers vs. non-smokers, which we discuss in section 4.4.
- Although I know that it is not the main focus of the article and the authors state that it is outside the scope of this review, I would suggest to shed light on CTTN in the formation of lung cancer. As a researcher in the field of thoracic cancer I am curious about the role of CTTN, therefore I would expand section 5.4.
Per the reviewer’s suggestion, we have significantly expanded section 5.4 to describe some of the more recent reports relevant to CTTN in lung cancer. In addition, Table 2 provides additional references and context regarding the possible roles of CTTN in lung cancer, and we also cite in the text several reviews exploring cancer.

Reviewer 2 Report
The manuscript entitled “Cortactin in Lung Cell Function and Disease” is a review paper focusing on the role of cortactin in lung physiology and pathology. Overall, this manuscript is well-written covering an understudied topic in a broad manner. Below are a few points that need to be addressed.
First, in Table 2 what is the link between CTTN and the various factors in lung cancer setting? An additional column could be added to describe in brief the particular association. miR-182 is not a protein.
Second, in vitro and in situ are usually written in italics (please see Figure 1 legend “InVitro”, “in situ”).
Author Response
Reviewer #2: The manuscript entitled “Cortactin in Lung Cell Function and Disease” is a review paper focusing on the role of cortactin in lung physiology and pathology. Overall, this manuscript is well-written covering an understudied topic in a broad manner. Below are a few points that need to be addressed.
- First, in Table 2 what is the link between CTTN and the various factors in lung cancer setting? An additional column could be added to describe in brief the particular association. miR-182 is not a protein.
We thank the reviewer for the kind comments about our work. We agree with the reviewer for raising an important point regarding linking CTTN with lung cancer, and we have substantially expanded our discussion in section 5.4, highlighting some recent studies on how cortactin expression and function may play a role in cell migration and invasion during lung cancer invasion and metastasis. We apologize for mislabeling Table 2 in a way that incorrectly suggests miR-182 is a protein. We have corrected the Table legend and text to indicate that Table 2 contains proteins “and other regulatory molecules” and signaling components.
- Second, in vitro and in situ are usually written in italics (please see Figure 1 legend “InVitro”, “in situ”).
We have revised the text to italicize these phrases. Thank you for your suggestion.
